# A Unique Anti-Cancer 3-Styrylchromone Suppresses Inflammatory Response via HMGB1-RAGE Signaling

**DOI:** 10.3390/medicines8040017

**Published:** 2021-03-24

**Authors:** Hideaki Abe, Miwa Okazawa, Takahiro Oyama, Hiroaki Yamazaki, Atsushi Yoshimori, Takanori Kamiya, Mitsutoshi Tsukimoto, Koichi Takao, Yoshiaki Sugita, Hiroshi Sakagami, Takehiko Abe, Sei-ichi Tanuma

**Affiliations:** 1Hinoki Shinyaku Co., Ltd., Chiyoda-ku, Tokyo 102-0084, Japan; hideaki.abe@hinoki.co.jp (H.A.); takahiro.oyama@hinoki.co.jp (T.O.); hymanami@gmail.com (H.Y.); T.Kamiya@hinoki.co.jp (T.K.); mugino.abe@hinoki.co.jp (T.A.); 2Department of Genomic Medicinal Science, Research Institute for Science and Technology, Organization for Research Advancement, Tokyo University of Science, Noda, Chiba 278-8510, Japan; miwa.okazawa@rs.tus.ac.jp; 3Institute for Theoretical Medicine Inc., Fujisawa, Kanagawa 251-0012, Japan; yoshimori@itmol.com; 4Department of Radiation Biosciences, Faculty of Pharmaceutical Sciences, Tokyo University of Science, Noda, Chiba 278-8510, Japan; tsukim@rs.tus.ac.jp; 5Department of Pharmaceutical Sciences, Faculty of Pharmacy and Pharmaceutical Sciences, Josai University, Sakado, Saitama 350-0295, Japan; ktakao@josai.ac.jp (K.T.); sugita@josai.ac.jp (Y.S.); 6Meikai University Research Institute of Odontology (M-RIO), Sakado, Saitama 350-0283, Japan; sakagami@dent.meikai.ac.jp

**Keywords:** inflammation, tumor microenvironment, carcinogenesis, RAGE, HMGB1, ERK 1/2, 3-styrylchromone

## Abstract

**Background:** High mobility group box 1 (HMGB1)-receptor for advanced glycation endo-products (RAGE) axis serves as a key player in linking inflammation and carcinogenesis. Recently, papaverine was revealed to suppress the HMGB1-RAGE inflammatory signaling pathway and cancer cell proliferation. Therefore, a dual suppressor targeting this axis is expected to become a new type of therapeutic agent to treat cancer. **Methods:** Papaverine 3D pharmacophore mimetic compounds were selected by the LigandScout software from our in-house, anti-cancer chemical library and assessed for their anti-inflammatory activities by a HMGB1-RAGE-mediated interleukin-6 production assay using macrophage-like RAW264.7 cells. Molecular-biological analyses, such as Western blotting, were performed to clarify the mechanism of action. **Results:** A unique 6-methoxy-3-hydroxy-styrylchromone was found to possess potent anti-inflammatory and anti-cancer activities via the suppression of the HMGB1-RAGE-extracellular signal-regulated kinase 1/2 signaling pathway. Furthermore, the 3D pharmacophore-activity relationship analyses revealed that the hydroxyl group at the C4′ position of the benzene ring in a 3-styryl moiety was significant in its dual suppressive effects. **Conclusions:** These findings indicated that this compound may provide a valuable scaffold for the development of a new type of anti-cancer drug possessing anti-inflammatory activity and as a tool for understanding the link between inflammation and carcinogenesis.

## 1. Introduction

A high correlation between chronic inflammation and carcinogenesis has been apparent from epidemiological and cellular-molecular analyses [1,2,3,4,5,6,7,8]. Various types of inflammation- and immune-related cells, such as macrophages, endothelial cells, natural killer (NK) cells, T cells, and myeloid-derived suppressor cells (MDSCs), join in a tumor and generate an inflammatory tumor microenvironment (TME) that impairs anti-cancer immunity and T cell tolerance and promotes cell proliferation, anti-apoptosis, survival, invasion, and metastasis (Figure 1). In particular, macrophages and MDSCs enter the cancer mass as the primary players in the inflammatory milieu of cancer [8,9,10,11,12]. The establishment of the TME depends on the activation of certain transcription factors, including nuclear factor-κB (NF-κB) and hypoxia-inducible factor-1α (HIF-1α), in cancer cells [13,14,15,16,17,18]. These transcription factors up-regulate the expression of inflammatory cytokines, such as interleukin-6 (IL-6) and tumor necrosis factor-α (TNF-α) [19,20,21,22]. These cytokines trigger the activation of inflammation as autocrines and paracrines, resulting in the release of pro-inflammatory mediators, such as high mobility group box 1 (HMGB1) and S100 family proteins (S100s), from macrophages and endothelial cells. These molecules, in turn, activate the receptor for advanced the glycation end-products (RAGE)-mediated signaling pathway and enhance inflammation in vicious feed-forward loops [23,24,25,26] (Figure 1). Subsequently, these RAGE ligands have been suggested to be involved in the establishment of TEM and become signals to activate carcinogenesis. However, the exact molecular and cellular mechanisms by which the HMGB1-RAGE signaling pathway participates in the establishment and maintenance of an inflammatory TEM, and consequently promotes carcinogenesis, remain to be solved.

Recently, it was discovered that papaverine (a non-narcotic, opium alkaloid medicine) and trimebutine (a spasmolytic drug) suppress the HMGB1-RAGE inflammatory signaling pathway and thereby attenuate sepsis [27,28]. Furthermore, it was demonstrated that papaverine can cancel the HMGB1-elicited proliferation of human glioblastoma cell lines, both temozolomide (TMZ)-sensitive U87MG and TMZ-resistant T98G cells [29]. In addition, the combination of papaverine and TMZ significantly delayed tumor growth in a U87MG xenograft mouse model [30]. RAGE, which is a receptor of the immunoglobulin superfamily that is primarily expressed in immune cells, is a multifunctional receptor that binds a broad repertoire of ligands, such as advanced glycation end-products (AGE), HMGB1, S100s, and Aβ, and mediates cellular responses to injury, infection, and stress conditions [31,32,33]. The expression of RAGE has been detected in a variety of human cancers, including brain, breast, colon, lung, prostate, ovary, and oral squamous cells, as well as lymphoma and melanoma [34,35]. In an environment of chronic inflammation leading to a pro-tumorigenic microenvironment (TME), RAGE ligands, including HMGB1, are accumulated and thereby modulate the TME and tissue dysfunction [33,36,37,38,39,40]. These observations support the notion of a direct link between HMGB1-RAGE-mediated inflammation and carcinogenesis. Therefore, these studies suggest that the HMGB1-RAGE axis can be used as a reliable therapeutic target against cancers.

As stated, the HMGB1-RAGE axis in TME is closely associated with carcinogenesis; thus, anti-cancer compounds targeting this axis that possess further anti-inflammatory activity may have far-reaching therapeutic potential. In this study, four 6-methoxy-3-styrylchromone (6M3SC) derivatives, which have already identified as anti-cancer agents [41,42,43,44], were selected by a papaverine 3D pharmacophore similarity search and examined in terms of their anti-inflammatory activities by a cell-based inflammatory assay method driven by the HMGB1-RAGE-mediated system using macrophage-like RAW264.7 cells. This research is the first to demonstrate a unique 6-methoxy-3-hydroxy-styrylchromone (compound **3**, 6M3HSC) that has both potent anti-cancer and anti-inflammatory activities. Furthermore, it suppressed downstream mediators of the HMGB1-RAGE signaling pathway, such as extracellular signal-regulated kinase 1/2 (ERK 1/2) activation, leading to inhibition of IL-6 production. In addition, this dual suppressive activity of 6M3HSC could be attributed to the hydroxyl group at the *para*-position of the benzene ring in a 3-styryl moiety. Thus, this compound may be a useful tool for understanding the functional role for HMGB1-RAGE signaling in linking inflammation to carcinogenesis and provide a novel scaffold for the development of a new type of anti-cancer drug possessing anti-inflammatory activity.

## 2. Materials and Methods

### 2.1. Chemicals and Reagents

Recombinant bovine HMGB1 was purchased from Chondrex Inc. (Redmond, WA, USA). Dulbecco’s Modified Eagle Medium (DMEM) and WST-8 reagent were purchased from Fujifilm Wako Pure Chemical (Osaka, Japan). Opti-MEM^®^ and Mouse IL-6 Uncoated ELISA Kit were purchased from Thermo Fisher Scientific (Waltham, MA, USA).

### 2.2. Synthesis of Test Compounds

(*E*)-6-methoxy-3-styryl-4*H*-chromen-4-one (compound **0**, 6M3SC), (*E*)-3-(4-fluorostyryl)-6-methoxy-4*H*-chromen-4-one (compound **1**), (*E*)-3-(4-(dimethylamino)styryl)-6-methoxy-4*H*-chromen-4-one (compound **2**), (*E*)-3-(4-(hydroxyl)styryl)-6-methoxy-4*H*-chromen-4-one (compound **3**), and (*E*)-3-(3,4-(hydroxyl)styryl)-6-methoxy-4*H*-chromen-4-one (compound **4**) (Figure 2) were synthesized by Knoevenagel condensation of the appropriate 3-formylchromone with selected phenylacetic acid derivatives, as described previously [42,44]. All the compounds were characterized by ^1^H NMR, MS spectra and elemental analyses after purification by silica gel column chromatography and recrystallization. All the compounds were dissolved in DMSO at 80 mM and stored at –20 °C before use.

### 2.3. Cells and Cell Culture

Mouse macrophage-like RAW264.7 cells were purchased from American Type Culture Collection (Manassas, VA, USA). RAW264.7 cells were cultured in DMEM supplemented with 10% fetal bovine serum (FBS) (Biosera Europe, Nuaillé, France) and 1% penicillin-streptomycin (Nacalai Tesque, Inc., Kyoto, Japan) in a humidified atmosphere containing 5% CO_2_ at 37 °C without antibiotics.

### 2.4. ELISA Assay for IL-6

RAW264.7 cells (6 × 10^4^ cells/dish) were seeded in a 96-well plate and incubated for 22 h. After changing the medium to Opti-MEM^®^, the RAW264.7 cells were pretreated with various concentrations of compounds for 2 h and stimulated with HMGB1. After 18 h of incubation, the concentrations of IL-6 in the culture supernatants were quantified using Mouse IL-6 Uncoated ELISA Kit, in accordance with the manufacturer’s instructions [27,28].

### 2.5. Assay for Cell Viability

After the removal of the supernatants for the IL-6 production assay in Section 2.4., WST-8 reagent containing DMEM-10% FBS (0.11 mL/well) was added to RAW264.7 cells and incubated for 30 min. After the incubation, the absorbance of the samples was detected at 450 nm using a Synergy™ HTX Multi-Mode Microplate Reader (BioTek Instruments, Inc.; Winooski, VT, USA). Cell viability was expressed as a percentage of the control cells treated with the vehicle alone RAW264.7 cells [27,28]. The same assay system was used to assess the viability of human glioblastoma T98G cells [29]. The treated human OSCC cell lines (Ca9-22, HSC-2, HSC-3, HSC-4) were incubated for 3 h in fresh medium containing 0.2 mg/mL of MTT reagent. Cells were then lysed with 0.1 mL of dimethyl sulfoxide and the absorbance at 546 nm of the cell lysate were determined using a microplate reader (Biochromatic Labsystem, Helsinki, Finland). The concentration of compound that reduced the cell viability by 50% (CC_50_) was determined from the dose–response curve [42,43,44].

### 2.6. Western Blot Analysis

RAW264.7 cells were lysed with RIPA buffer (NACALAI TESQUE, INC., Kyoto, Japan) containing a protease and phosphatase inhibitor cocktail (Roche, Basel, Switzerland). The lysates were centrifuged at 10,000× *g* for 10 min at 4 °C. The supernatants were used as cell extracts for immunoblotting analysis. Total proteins were measured by the TaKaRa BCA Protein Assay Kit (Takara Bio Inc., Kusatsu, Japan) according to the manufacturer’s instructions. The supernatant fractions were mixed with 6× loading dye (NACALAI TESQUE, INC., Kyoto, Japan) and then samples were applied to a 12.5% Extra PAGE One Precast Gel (NACALAI TESQUE, INC., Kyoto, Japan). Electrophoresis was performed at a constant current of 20 mA for 90 min in running buffer (NACALAI TESQUE, INC., Kyoto, Japan). Proteins were transferred to a polyvinylidene fluoride (PVDF) membrane (GE Healthcare, Chicago, IL, USA) at a constant voltage of 50 V for 120 min. The primary and secondary antibody reactions were performed with an iBind™ Flex Western System (Thermo Fisher Scientific Inc., Waltham, MA, USA). The total and phospho-ERK1/2 proteins were detected using their rabbit antibodies (Cell Signaling Technology Inc., Danvers, MA, USA) followed by anti-rabbit HRP-labeled IgG secondary antibody (GE Healthcare). The GAPDH protein was detected using rabbit antibodies (Proteintech Group, Inc., Rosemont, IL, USA), followed by anti-rabbit HRP-labeled IgG secondary antibody (GE Healthcare)., Chemiluminescent reactions were performed using Immobilon® ECL Ultra (Merck KGaA, Darmstadt, Germany) and signals were detected by an iBright CL1000 system (Thermo Fisher Scientific Inc., Waltham, MA, USA).

### 2.7. In Silico 3D Pharmacophore Analysis

In silico 3D pharmacophore similarity analyses were performed with LigandScout software [45]. All compounds were constructed by Chem-Draw Professional 18.0 and converted to 3D conformations by the optimization program in LigandScout software [46,47].

### 2.8. Statistical Analysis

The numbers of biological and statistical significance are presented in the figure legends. Data are expressed as mean ± standard error (SE). SE is defined as the average of standard deviation in more than three independent experiments. Statistical analyses were performed using the Microsoft^®^ Excel^®^ software. Immunoblot data analyses were conducted using one-way ANOVA followed by Dunnett’s test for multiple comparisons or by the Student t-test for comparison between two variables. *p* values < 0.05 were considered statistically significant.

## 3. Results

### 3.1. Anti-Inflammatory Activities of 6M3SC Derivatives Assessed by Suppression of IL-6 Production in HMGB1-Stimulated RAW264.7 Cells

In this study, four 6M3SC derivatives (compounds **1**, **2**, **3** and **4**), which have potent anti-cancer activities (CC_50_ < 50 μM), and one basic 6M3SC compound **0** (Figure 2) [42,44], were selected by 3D pharmacophore similarity analyses to papaverine (3DPFS > 0.4) (Table 1), and examined for their anti-inflammatory activities by HMGB1-stimulated IL-6 production assay using macrophage-like RAW264.7 cells. Figure 3a depicts the representative titration curves of the suppression of IL-6 production by the treatment of these 6M3SC derivatives. The 50% effective concentration (EC_50_) values of compounds **0**, **1**, **2**, **3,** and **4** were calculated to be > 100, > 100, 24.6, 0.67, and 1.0 μM, respectively, from the titration curves. Of the compounds tested, compound **3** (6-methoxy-3-hydroxy-styrylchromone, 6M3HSC) had the most potent suppressive effect on IL-6 production in HMGB1-stimulated RAW 264.7 cells. The suppressive activity of compound **3** was considerably higher than that of the dimethylamine-substituted compound **2**. The *para*- and *meta*-dihydroxyl-substituted compound **4** had a fairly low suppressive activity compared with compound **3**. Compounds **0** and **1** had a little suppressive effect on IL-6 production. Additionally, these 6M3SC derivatives showed no obvious cytotoxic effects on RAW 264.7 cells at concentrations up to 30 μM (Figure 3b). These results indicated that the substituent of the hydroxyl group at the *para*-position of the benzene ring in a 3-styryl moiety in compound **3** is significant in the attenuation of HMGB1-stimulated, pro-inflammatory response in macrophages.

### 3.2. Compound 3 Suppresses HMGB1-RAGE Signaling by Inhibiting the Activation of ERK 1/2

As noted above, compound **3** (6M3HSC) strongly suppressed the HMGB1-stimulated IL-6 production like papaverine, which inhibits the HMGB1-RAGE axis. Therefore, we focused on the molecules downstream of the RAGE signaling pathway, which are affected by the treatment of compound **3**. Through the binding of HMGB1 to RAGE, the cytoplasmic tail of RAGE has been reported to interact with key mediator kinases ERK 1/2, leading to the activation of downstream RAGE effectors, such as NF-κB and AP-1 [48], the influence of compound **3** on the phosphorylation of ERK 1/2 was examined (Figure 4, Appendix A). The level of phosphorylation of ERK 1/2 was determined by Western blotting. Compared with that of the HMGB1-non-treated control group, the phosphorylation levels of ERK 1/2 were obviously increased in the HMGB1-stimulated group. Importantly, in the HMGB1-stimulated group compound **3** extensively suppressed the increased phosphorylation levels of ERK 1/2 in a dose-dependent manner (Figure 4). The results indicated that the activation of ERK 1/2 by interaction with the extracellular tail of RAGE was considerably inhibited by compound **3**, resulting in a marked decrease in the production of inflammatory cytokines, such as IL-6 (Figure 3a).

### 3.3. Structural Insights into 6M3SC Derivatives for Designing Novel Dual Anti-Inflammatory and Anti-Cancer Agents

The aforementioned observations indicated that the substituents at the *para*-position were electrostatically important in the anti-inflammatory and anti-cancer activities. As such, the four compounds were evaluated to gain further insight into the effects of the substituents. The 3D pharmacophore analyses by LigandScout are useful tools to characterize the 3D electrostatic features of small molecules (Figure 5). As summarized in Table 2, the apparent differences between the four compounds are the numbers of hydrophobic regions (H), hydrogen bond acceptors (HBA), and donors (HBD). All compounds have the same aromatic regions. Compounds **3** and **4** contain a hydrogen bond donor at the *para*-position of the 3-styryl moiety. Compound **4** had a more hydroxyl group at the *meta*-position and less anti-inflammatory activity as compared with compound **3** (Figure 3a). On the other hand, when the hydroxyl group of compound **3** was substituted for the hydrophobic dimethylamino group (compound **2**), the anti-inflammatory activity was considerably lower (Figure 3a).

As compound **3**, whose C4′ position is the hydroxyl group, had the highest anti-cancer activity (Table 1), the functional hydroxyl group at the *para*-position of the benzene ring in the 3-styryl moiety may be necessary to donate anti-cancer activity to the 6M3SC. In contrast, the hydrophobic dimethylamino group (compound **2**) was revealed to be more beneficial for anti-cancer activity than the di-hydroxyl group (compound **4**) (Table 1). These results suggest that the structural improvements and optimization of the *para*-position of the benzene ring in the 3-styryl moiety may be important to generate a new type of anti-cancer agent that possesses potent anti-inflammatory activity. Furthermore, additional structural improvements to compound **3,** via the attachment of appropriate side chains at the proximate *meta*-/*para*-positions of the chromone moiety and *meta*-/*ortho*-positions of the 3-styryl moiety, might increase both anti-cancer and anti-inflammatory activities and ameliorate the issue. As such, a new dual anti-cancer scaffold with substituents at these positions should be designed from the seed structure of compound **3** (Figure 6).

## 4. Discussion

Accumulating evidence has suggested the importance of the interactions of RAGE and RAGE ligands, such as HMGB1, S100s, AGE, and Aβ, in the pathogenesis of various human diseases, including cancer (Figure 7) [49,50,51,52,53,54,55,56,57]. RAGE-RAGE ligand signaling triggers the activation of its downstream mediators/activators, such as ERK ½; diaphanous-related Formin-1 (DIAPH1); dedicator of cytokinesis 7 (DOCK7); Toll/interleukin-1 receptor domain-containing adaptor protein (TIRAP); myeloid differentiation primary response gene 88 (MyD88); interleukin-1 receptor-associated kinase 1, 2, 4 (IRAK1/2/4); Rho GTPase, Ras-related C3 botulinum toxin substrate 1 (Rac1); cell division cycle 42 (Cdc42); AKT; c-Jun N-terminal kinase (JNK); TGF-β-activated kinase 1 (TAK1); nuclear factor kappa-light-chain-enhancer of activated B cells (NF-κB); and activator protein 1 (AP-1) (Figure 7) [48,58,59,60,61,62,63]. These protein factors have been reported to play significant roles in regulating inflammatory signaling in innate immune responses and promoting cancer cell growth and metastasis in various cancers (Figure 1) [29,30,56,62]. Furthermore, in numerous cancers chronic inflammation via the HMGB1-RAGE signaling pathway has been shown to change their TME and stimulate the development of cancer. Thus, the HMGB1-RAGE axis serves as a key player in linking inflammation and carcinogenesis. Therefore, dual inhibitors possessing anti-cancer and anti-inflammatory activities may become effective and rational anti-cancer drugs.

In this study, we identified compound **3** as a dual anti-cancer and anti-inflammatory molecule from 6M3SC derivatives. Although small molecular candidates for the suppression of the RAGE-mediated signaling pathway have been reported, none have been approved for clinical use to date [27,28,63]. The use of these candidates in clinical applications for cancer chemotherapy requires knowledge of the mechanism of action in signaling pathways and the gene regulation of both inflammation and carcinogenesis (Figure 7). To understand the suppressive effect of compound **3** on the HMGB1-RAGE inflammatory signaling pathway, we investigated the underlying mechanism of action in the macrophage-like RAW264.7 cells. It is noted that this compound strongly inhibits the activation of ERK 1/2 (Figure 4), which are important mediators recruiting to the tail domain of RAGE [48], resulting in the suppression of HMGB1-mediated pro-inflammatory responses. Thus, the treatment of compound **3** resulted in the considerable suppression of the production of pro-inflammatory cytokines, such as IL-6. The suppression effect of this compound on the HMGB1-RAGE-ERK 1/2 signaling pathway may be involved in blocking the generation of TEM, anti-cancer activity, and exerting protective effects against cancer.

Our finding of the dual inhibitory activities of compound **3** on both anti-cancer and anti-inflammation may provide a new means to further develop novel, effective, and safe anti-cancer drugs. Importantly, the 3D pharmacophore-activity relationship analyses revealed that the *para*-position (C4′) of the benzene ring in the 3-styryl moiety plays a significant role in both anti-inflammatory and anti-cancer activities. Indeed, compound **3**, which has a hydroxyl group of hydrogen bond donors at the C4′ position, had the highest anti-inflammatory and anti-cancer activities. Thus, the optimization of the *para*-position of the benzene ring in the 3-styryl moiety may be important in the development of more effective dual anti-cancer pharmaceuticals. Furthermore, additional structural improvements to this compound via the attachment of appropriate side chains at the proximate *meta*- and *ortho*- positions should increase the anti-cancer activity and ameliorate the issue. As such, a novel dual anti-cancer scaffold compound with substituents at the *para*-, *meta*-, and *ortho*-positions of the benzene ring in the 3-styryl moiety as well as the heterocyclic chromone moiety was designed (Figure 6).

Further studies on the dual-active agents possessing anti-cancer and anti-inflammatory activities using rigid synthesized derivatives will allow a more definitive understanding of the unique pharmacophore of 6M3HSC (compound **3**). In addition, it is imperative to determine the target(s) of the dual-active agents. It is worth considering whether they inhibit key targets regulating anti-cancer and anti-inflammatory activities. This is an important subject of ongoing investigation. Although additional confirmatory studies, such as the synthesis of novel 6M3HSC derivatives and details of in vivo efficacy as well as in vitro activity are warranted, the results of this research provide invaluable insight into not only the beneficial design of effective anti-cancer drugs that offer improved therapeutic potential but also the linking of inflammation to carcinogenesis.

## 5. Conclusions

The RAGE-RAGE ligand signaling pathway has been shown to drive the establishment of TME and is an important target for cancer therapeutics currently under development [29,30,39,63,64]. This research identified a unique 6M3HSC derivative, compound **3**, which possesses anti-cancer activity, as a potent anti-inflammatory compound by blocking the RAGE-HMGB1 signaling pathway via ERK 1/2 activation. Furthermore, the present study demonstrated that the *para*-position of benzene ring in the 3-styryl moiety may play a significant role in the generation of effective dual pharmaceuticals that have both anti-cancer and anti-inflammatory activities for cancer chemotherapies. Such dual anti-cancer agents could be effective in dissolving TEM, preventing cancer progression, metastasis, and immune evasion. Furthermore, they provide a viable therapeutic strategy for treating almost all cancers. However, as RAGE-RAGE multi-ligand signaling systems have also been shown to be involved in cellular repair and tissue regeneration as well as pro-carcinogenic inflammation (Figure 7), potential side effects must also be considered in the therapeutic use of RAGE inhibitors [33,65,66,67]. Therefore, genome-wide analyses will be required to derive a comprehensive understanding of the RAGE signaling pathway and response for the development of effective cancer therapeutics. Nevertheless, encouraged by the findings of the present study, further optimizations of 6M3HSC to generate enhanced dual anti-cancer lead compounds possessing more potent anti-inflammatory activities are currently being conducted.

## Figures and Tables

**Figure 1 medicines-08-00017-f001:**
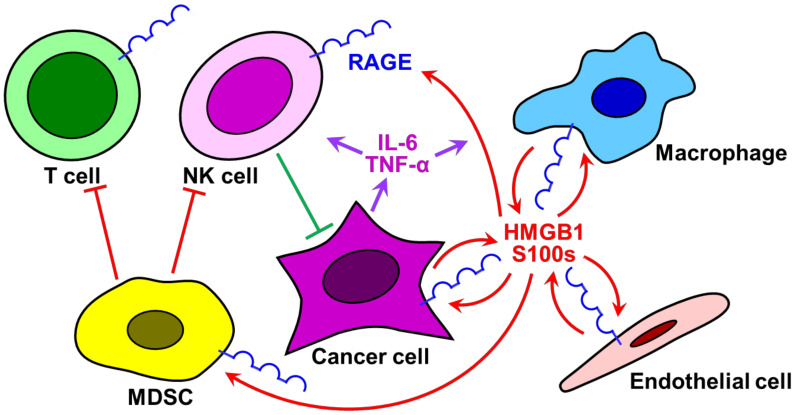
Schematic illustrating the functional role of receptor for the advanced glycation endo-products (RAGE)-RAGE ligand signaling in linking inflammation to carcinogenesis. In a microenvironment of tumor, where cancer cells up-regulate NF-κB expression by hypoxia-inducible factor-1α (HIF-1α), an inflammatory tumor microenvironment (TME) is established by the secretion of pro-inflammatory cytokines, such as IL-6 and TNF-α, and thereby the release of RAGE ligands, such as HMGB1 and S100s, from recruited macrophages and endothelial cells. The RAGE-RAGE ligand signaling activates inflammation in feedforward loops, resulting in the recruitment of NK and T cells, and further myeloid-derived suppressor cells (MDSCs). MDSCs suppress NK and T cells, leading to T cell tolerance and impaired anti-cancer immunity.

**Figure 2 medicines-08-00017-f002:**
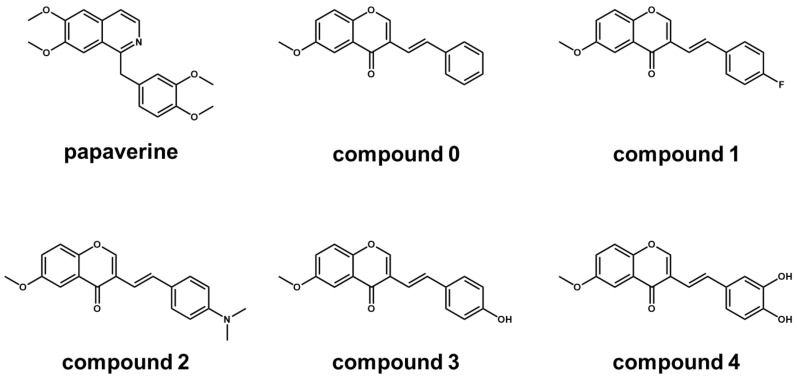
Structures of papaverine and five 6M3SC derivatives.

**Figure 3 medicines-08-00017-f003:**
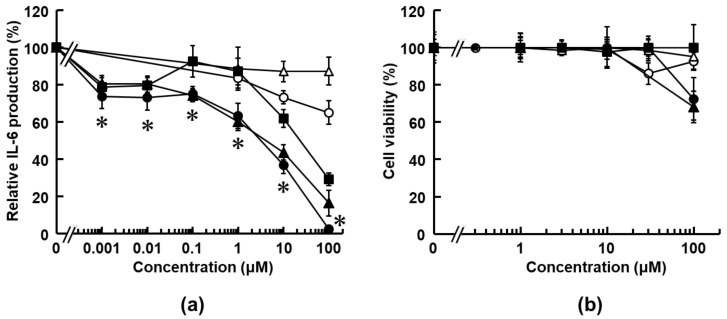
Anti-inflammatory activities of 6M3SC derivatives in HMGB1-stimulated RAW264.7 cells. (**a**) The cells were treated with indicated concentrations of each compound: **0** (open circle), **1** (open triangle), **2** (closed square), **3** (closed circle), and **4** (closed triangle). The IL-6 production (% of control) was measured using the IL-6 ELISA system, as described in the Materials and Methods section. Values are mean ± standard error (SE) for four independent experiments, and the bars indicate the SE values. (**b**) Cell viability was determined by the WST-8 assay, as described in the Material and Methods section. The data are presented as mean ± SE of four independent experiments and * *p* < 0.05 was accepted as a significant difference compared with untreated samples for compound **3**.

**Figure 4 medicines-08-00017-f004:**
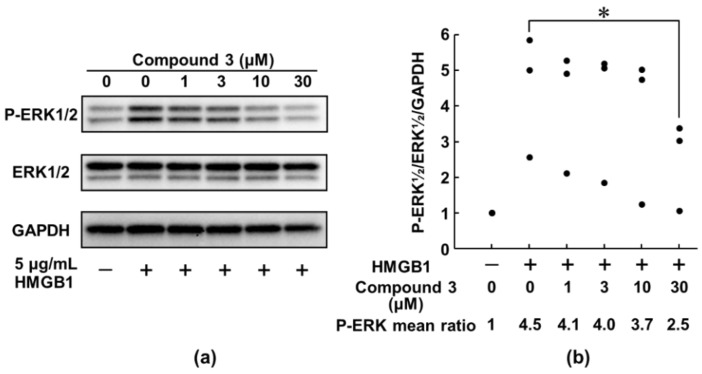
Suppressive effect of compound **3** on the HMGB1-induced activation of ERK 1/2. RAW264.7 cells were pretreated with compound **3** at the indicated concentrations for 2 h and then stimulated with HMGB1 (5 μg/mL) for 18 h. Cell lysates were subjected to Western blotting using phospho-ERK 1/2 (P-ERK 1/2) and total ERK 1/2 antibodies as described in the Materials and Methods section. GAPDH served as a loading control. The panels depicted a typical immunoblot (**a**) and values presented as mean ratio of three independent experiments of each P-ERK 1/2, ERK 1/2, and GAPDH band intensities were calculated by densitometric analysis (**b**). * *p* < 0.05 was accepted as a significant difference.

**Figure 5 medicines-08-00017-f005:**
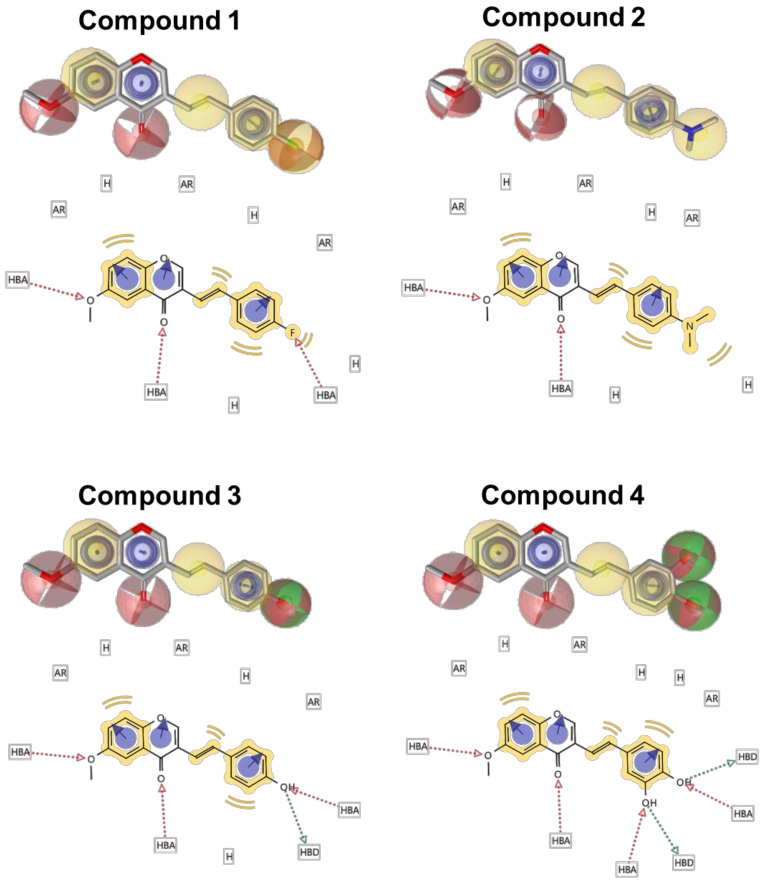
Structural characteristics of four 6M3SC derivatives analyzed by the LigandScout software. The 3D pharmacophore models and 2D characteristics of four 6M3SC compounds were analyzed by the LigandScout software. AR, Aromatic; H, hydrophobic; HBA, hydrogen-bond acceptor; HBD, hydrogen-bond donor. Oxygen and nitrogen atoms are illustrated with red and blue sticks, respectively, and the carbon atoms of these compounds are depicted by gray sticks. Aromatic rings, double bonds, and methoxy groups are illustrated with blue circles, yellow, and green, respectively.

**Figure 6 medicines-08-00017-f006:**
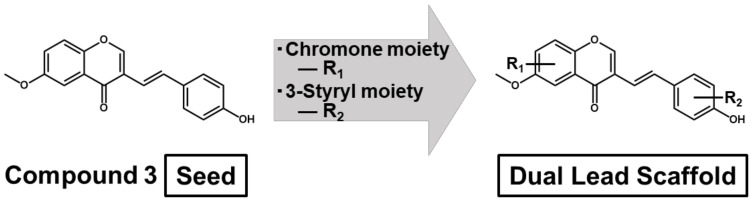
Designing a new lead scaffold for the development of novel dual pharmaceuticals possessing anti-cancer and anti-inflammatory activities. The two sections of the 3-styrylchromone molecule, the heterocyclic chromone, and 3-styryl benzene ring moieties are subjected to structure optimization studies for functional groups of R1 and R2, respectively.

**Figure 7 medicines-08-00017-f007:**
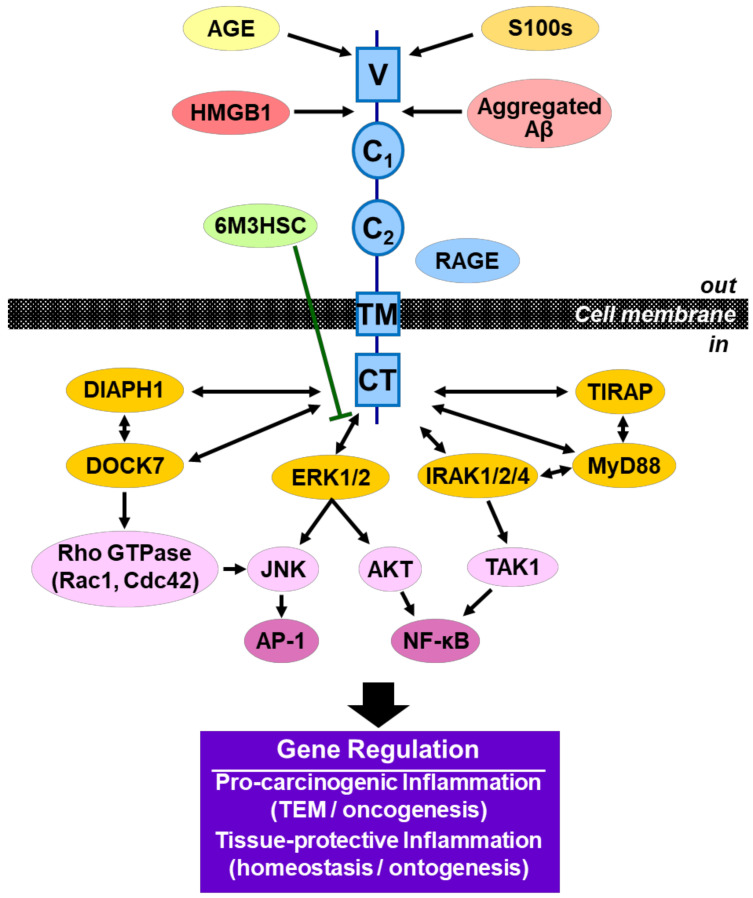
Schematic illustrating the suppressive effect of 6M3HSC (compound **3**) against the RAGE-RAGE ligand signaling pathway. A potential mechanism of the suppressive effect of this compound **3** on HMGB1-RAGE-ERK 1/2 signaling pathway is shown. The interactions of multi-ligands with RAGE up-regulate the RAGE downstream signaling pathway. Consequently, RAGE-regulated genetic programs of pro-inflammation and carcinogenesis (TEM and oncogenesis) or cell repair and tissue regeneration (homeostasis and ontogenesis) start driving by the action of transcription factors, such as NF-κB and AP-1. RAGE structure includes the variable (V), constant (C_1_ and C_2_) domains, the transmembrane region (TM), and the cytoplasmic tail (CT).

**Table 1 medicines-08-00017-t001:** Anti-cancer activities of five 6M3SC derivatives and their 3D pharmacophore similarities against papaverine.

Compound	Anti-Cancer Activity CC_50_ (μM) ^a^	3D Pharmacophore Similarity 3DPFS ^b^
**papaverine**	40.0	1
**0**	>50	0.51
**1**	49.0	0.51
**2**	6.5	0.42
**3**	2.0	0.42
**4**	16.0	0.42

^a^ The anti-cancer activity was assayed as described previously [29,42,43,44]. The CC_50_ value of anti-cancer activity was determined from the dose–response curve, as described in the Materials and Methods section, using T98G cells for papaverine [29] and human OSCC cell lines Ca9-22, HSC-2, HSC-3, and HSC-4 for **0**, **1**, **2**, **3,** and **4** [42,43,44]. Values are means for four independent experiments. ^b^ The 3D pharmacophore fit scores (3DPFS) were analyzed by LigandScout software, as described in the Materials and Methods section.

**Table 2 medicines-08-00017-t002:** Characteristics of the functional groups of the four 6M3SC derivatives.

Compound	Number			
	AR ^a^	H ^b^	HBA ^c^	HBD ^d^
**1**	3	4	3	0
**2**	3	4	2	0
**3**	3	3	3	1
**4**	3	3	4	2

Characteristics of the compounds were analyzed by LigandScout, as described in the Materials and Methods section. ^a^ Aromatic region; ^b^ hydrophobic region; ^c^ H-bond acceptor; ^d^ H-bond donor.

## Data Availability

Not applicable.

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
