# Peer review of "A Unique Anti-Cancer 3-Styrylchromone Suppresses Inflammatory Response via HMGB1-RAGE Signaling"

_medicines, 2021, doi:10.3390/medicines8040017_

Round 1

Reviewer 1 Report

Although the findings that 6M3SC derivatives were found to have potent anti-inflammatory and anti-cancer activities via the suppression of HMGB1-RAGE-ERK1/2 signaling pathway are interesting, a number of points need clarifying and certain statements require further justification. These are given below.

<Major points>

  1. The authors used only a mouse macrophage-like RAW264.7 cell line. Although the results are observed in RAW264.7 cells, whether similar results are obtained in other cells is not certain. The authors should test other cells than 7 cells.
  2. Numbers of experiments should be described in Figures 3 and 4.
  3. There is no error bar in Fig. 4(b).
  4. Figure 1 should be “graphic abstract”.

<Minor points>

  1. Line 80: ERK should be abbreviated.
  2. Line 90: The font of “and WST-8 reagent” should be changed to “Palatino Linotype”.
  3. Line 98: “Figure 1” should be changed to “Figure 2”.
  4. Line 128: “Shiga” should be changed to “Kusatsu”.
  5. Line 134: “IL, USA” should be changed to “Chicago, IL”.
  6. Line 136, “MA, USA” should be changed to “Waltham, MA”.
  7. Line 138: “MA, USA” should be changed to “Danvers, MA”.
  8. Line 140: “IL, USA” should be changed to “Rosemont, IL”.
  9. Line 199: “interact with” should be changed to “interact with”.
  10. Line 276: “TME” has already abbreviated in Line 28 (tumor microenvironment (TME)). Please be consistent.

Line 294: “ant-cancer” should be changed to “anti-cancer”.

Author Response

Dear Reviewer 1,

We greatly appreciate your critical reading of our manuscript and kind suggestions to this study. According to your suggestions, we have modified the manuscript as follows.

Major points

  1. I feel that this comment is not relevant suggestion in this paper, although in a way, the comment is right. Because the main point of this study, for the first time, is the achievement of a scaffold hopping by our in silico analyses from the 3D pharmacophore of papaverine to a unique 3-styrylchromone derivative, 6-methoxy-3-(4-hydroxy)-styrylchromone (compound 3) and the finding that compound 3 has both potent anti-cancer and anti-inflammatory activities via the suppression of HMGB1-RAGE-ERK 1/2 signaling pathway. In addition, 3D pharmacophore-activity relationship analyses revealed that the hydroxyl group at the C4' position of the benzene ring in a 3-styryl moiety was significant in the dual suppressive effects. These findings indicate that this compound may provide a valuable scaffold of the development of a new type of anti-cancer drug possessing anti-inflammatory activity. So, we would like to publish the detail data using not only other macrophage-like cell lines but also normal native macrophages and RAGE-KO/KD macrophages as an another research article elsewhere.
  2. The numbers of experiments in Figure 3 data had been mentioned in its legend in the original manuscript. We apologize for not including the mention about the numbers of experiments in Figure 4. We have performed the statistical analysis of Figure 4a. This result has been shown in new Figure 4b in the revised manuscript. We would like to publish the detailed analysis of the downstream mediators/effecters via HMGB1-RAGE-ERK 1/2 signaling pathway as an another research article elsewhere.
  3. According to your suggestion, Figure 4 and its legend have been improved in new Figure 4 in the revised manuscript.
  4. We have made Graphical Abstract using Figure 1.

Minor points

All mistakes raised in Minor points have been corrected in the revised manuscript.

Reviewer 2 Report

The authors describe a new small molecule that shows anti-inflammatory activity and anticancer activity and could serve as lead for generation of additional small molecule inhibitors.

Major concerns

  1. The authors have not investigated in detail the effect of the compound on HMGB1-RAGE signaling, they only showed the effect in pERK, but not if this has an impact on additional downstream pathways.
  2. The mechanism of action of this compound is not described, i.e., how does it affect signaling? Also, single measurement of IL6 is not enough to support such a strong statement that the compound has anti-inflammatory activity. Other cytokines or an in vitro cell study should be evaluated.

Minor concerns

  1. Describe briefly the method used to assay anti-cancer activity in the methods sections, even if citations are provided.
  2. Fig. 3- conduct statistical analysis to show significance of the findings.
  3. Fig. 4 - indicate if the western blot is representative of how many experiments. In the graph, please, represent single replicates and results as mean -/+ SD.

Author Response

Dear Reviewer 2,

We greatly appreciate your kind productive suggestions of our work.

Major concerns

  1. We would like to publish the detailed analysis of the downstream mediators/effecters via HMGB1-RAGE-ERK 1/2 signaling pathway as an another research article elsewhere.
  2. According to the suggestion, we have included the data of the effect of compound 3 on TNF-α production. The result has been mentioned in the Results section of 3.1. in the revised manuscript. Also, the method of ELISA assay for TNF-α has been added in the Materials and Methods section 2.4. in the revised manuscript.

Minor concerns

  1. We apologize for not including sufficient description of the methods using assay for anti-cancer activity in the Materials and Methods section. The methods have been incorporated into the Materials and Methods section of 2.5. in the revised manuscript.
  2. As mentioned in Discussion section, further studies are required to elucidate the molecular mechanisms by which compound 3 exerts both anti-cancer and anti-inflammatory activities and identify its target(s). These are important subject of ongoing investigation. We would like to publish these data and the suppressive effect on the production of cytokines and chemokines as an another research article elsewhere. Furthermore, the main point of this study, for the first time, is the achievement of a scaffold hopping by our in silico analyses from the 3D pharmacophore of papaverine to a unique 3-styrylchromone derivative, 6-methoxy-3-(4-hydroxy)-styrylchromone (compound 3) and the finding that compound 3 has both potent anti-cancer and anti-inflammatory activities via the suppression of HMGB1-RAGE-ERK 1/2 signaling pathway. In addition, 3D pharmacophore-activity relationship analyses revealed that the hydroxyl group at the C4' position of the benzene ring in a 3-styryl moiety was significant in the dual suppressive effects. These findings indicate that this compound may provide a valuable scaffold of the development of a new type of anti-cancer drug possessing anti-inflammatory activity.
  3. According to the suggestion, numbers of experiments and mean ± SE have been mentioned in the legend of Figure 4 in the revised manuscript.

Reviewer 3 Report

High mobility group box 1 (HMGB1)-receptor for advanced glycation endoproducts (RAGE) axis plays an important role in inflammation and carcinogenesis. In this paper, papaverine 3D phrmacophore mimetic compounds were selected to test their anti-cancer as well as anti-inflammatory effects. Among several candidate small molecules, compound 3 showed the strongest anti-cancer activity by suppressing IL-6 production and ERK1/2 phosphorylation under the stimulation of HMGB1. Also, it has been suggested that compound 3 can be improved as a new lead scaffold by adding chromone moiety and 3-styryl moiety. The results are shown very clearly with supportive data. So, the paper is considered to reach a sufficient level to be accepted for publication. The only point is that line 157 should be improved as follows:

From “four 6M3SC derivatives (compound 0, 1, 2, 3 and 4), which …”

To “four 6M3SC derivatives (compound 1, 2, 3 and 4), which …” (0 should be removed)

Author Response

Dear Reviewer 3,

We greatly appreciate your kind suggestion of our study.

According to the suggestion, 0 has removed from the sentence of “four 6M3SC derivatives (compound 0, 1, 2, 3 and 4)” in the revised manuscript (P 4, line 183, it may be out of position by display mode of MS Word).

Round 2

Reviewer 1 Report

Judged by the revised version (medicines-1134772-v2) and responses from the authors, all the points are suitably revised in the R2 version.

Author Response

Dear Reviewer 1,

Thank you for your kind review of our manuscript. We have done our best to modify the manuscript according to your suggestions. This is the first report concerning the structural conversion of papaverine to a unique anti-cancer 3-styrylchromone derivative, 6-methoxy-3-(4-hyroroxy)-styrylchromone (Compound 3), and the discovery of its potent anti-inflammatory activity via the suppression of HMGB1-RAGE-ERK1/2 signaling pathway. We will further expand this study to develop a new types of anti-cancer drugs.

Thank you again for your best consideration for the revised manuscript.

Reviewer 2 Report

The authors have partially addressed my concerns; however, the authors need to introduce some modifications before the manuscript can be accepted for publication.

Major concerns

1. There is a duplication of panels in figure 3, and I could not find any reference to TNFalpha in section 3, as the authors claimed in their point-by-point reply.

2. Similarly, there is a duplication in Figure 4, besides showing two different ways of quantifying the results. The authors need to correct the figure and provide original scans of their immunoblots. Please represent the relative pERK or ERK signal (top b panel) for each independent replicate in the graphs. For figure 4, bottom panel b, please, represent the relative pERK level for each independent replicate in the chart.

Author Response

Dear Reviewer 2,

In spite of your critical reading of our manuscript and kind suggestions, we have replied something weird. We apologize for the confusing responses against the Major concerns No.2, which was occurred by the wrong position of this response due to exchanging of the Minor concerns No.2. Also, in order to show you the original Figures 3 and 4, we have left old ones with amended new ones, that is the reason of the duplication of panels in Figures 3 and 4. ( In Figure 2, an old original panel has been also left as a duplication one.) These mistakes have been corrected in the 2nd revised manuscript.

Major concerns

  1. We are now examining the molecular mechanisms by which compound 3 exerts both anti-inflammatory and anti-cancer activities and identifying the molecular target(s) of compound 3 in relation to the suppression effect on the production of various cytokines, including TNF-α and IL-1β, and chemokines, such as CCL2 and CXCL2, and their gene regulation by NF-κB, AP-1 and STAT1. Furthermore, main point of this study, for the first time, is the in silico conversion of papaverine to unique 3-styrylchromone derivatives, especially to 6-methoxy-3-(4-hydroxyl)-styrylchromone (compound 3) using LigandScout software, and the finding that compound 3 has both potent anti-cancer and anti-inflammatory activities via the suppression of HMGB1-RAGE-ERK 1/2 signaling pathway. So, we would like to publish these data mentioned above as an another research article elsewhere. We apologize for leaving the old Figure 3 and putting a confused duplication of panels in Figure 3. The old original Figure 3 (top panel) has been deleted and only new Figure 3 has been uploaded in the revised manuscript.

  1. We also apologize for including the old Figure 4, resulting in a duplication of panels in Figure 4. The old original Figure 4 (top panel) has been deleted and only new Figure 4 has been uploaded. The relative P-ERK levels have been represented in new Figure 4 in the revised manuscript. (Also, the old Figure 2 has been deleted and only new one has been put in the revised manuscript.)

Round 3

Reviewer 2 Report

1. There is still a duplication in Figure 4, otherwise, label the panels as a through d. Still, represent the relative pERK value for each replicate in the graph in addition to the mean -/+ SD, instead of a bar.

2. Please, provide the original scans for the immunoblots, as requested in my previous report.

Author Response

Dear Reviewer 2,

Thank you for your kind review of our revised manuscript. We have done our best to modify the manuscript.

  1. In order to able to compare new amended Figure 4 with old one, a duplication of Figure 4 was prepared in the revised manuscript by the Editorial office of Medicines. We had sent only a single new amended Figure 4 in the revised manuscript.

  1. We do not understand why the statement “Please provide the original scans for the immunoblots,” was made. As mentioned above, a confusing duplication of Figure 4 was prepared by the Editorial office; that is, one is a new amended Figure and the other is its old original one. We had already provided the original scans of the immunoblot and displayed as relative P-ERK level (ratio) in the new Figure 4b in the revised manuscript.

Round 4

Reviewer 2 Report

I appreciate the author's efforts to address my concerns.

1. With regard to the author's comment: "We do not understand why the statement “Please provide the original scans for the immunoblots,” was made"; let me clarify that it is common practice in many journals to display the original scans for immunoblots and show where the scan was cropped, for example, as a supplemental figure.

2. In addition, since the experiment in figure 4 has been conducted three times, as I have requested before, I would like the authors to represent the relative value for pERK for each replicate (i.e., one dot for each replicate's value) instead of a bar -/+ SD.

Author Response

Dear Reviewer 2,

We appreciate your suggestions of our work. Our responses to the comments as follows:

  1. According to your suggestion, we have provided the original scans for the immunoblots, where the scans were not cropped, as a supplemental Figure 1.

  1. According to your suggestion, we have modified Figure 4b, one dot for each replicate’s value for P-ERK.
